# The Influence of Repair Quality on Aircraft Spare Part Demand Variability

Lars M. Heijenrath [1] and Wim J. C. Verhagen [2,*]

1 Section Air Transport & Operations, Department Control and Operations, Faculty of Aerospace Engineering, Delft University of Technology, 2629 HS Delft, The Netherlands
2 Aerospace Engineering & Aviation, School of Engineering, RMIT University, Melbourne 3000, Australia
* Correspondence: wim.verhagen@rmit.edu.au

**Abstract:** Accurate estimation of spare part demand is challenging in the case of intermittent or lumpy demand, characterised by infrequent demand occurrence and variability in demand size. While prior research has considered the effect of exogenous variables on spare part demand, there is a lack of research considering the effects of repair quality and aggregated spare part demand behaviour across fleets of assets under the influence of multiple simultaneously acting drivers of failure. This research provides new insights towards the problem of estimating variable spare part demand through modelling and simulation of the effects of multiple, simultaneously considered spare part demand drivers. In particular, a contribution to the state of the art is introduced by the use of a Branching Poisson Process (BPP) to model repair quality effects for spare part demand generation in conjunction with several demand drivers. The approach is applied in a numerical study which involves component failure characteristics based on real-life data from an aircraft maintenance, repair and overhaul (MRO) provider. It is shown that repair quality improvements drive down the variance in the demand and the total number of failures over time, and outperform the effect of environmental drivers of failure in terms of demand generation.

**Keywords:** spare part demand; repair quality; aircraft components; Branching Poisson Process; intermittent demand

## 1. Introduction

In aviation, it is crucial for airlines to maintain their fleet in an airworthy state. Each individual aircraft is required to meet a high standard of technical reliability, which is accomplished through maintenance. Aircraft maintenance encompasses a variety of tasks that can be deployed to keep the aircraft in an airworthy state. One of these tasks is replacement. As components are replaced, demand for new components is generated. To minimize the associated downtime of an aircraft, maintenance, repair and overhaul (MRO) providers (a term used here to indicate the contributions of Part 145 Approved Maintenance Organizations and Part M Continued Airworthiness Maintenance Organizations) aim to meet this generated demand by having spare parts available in their inventory. However, available inventory may not always be sufficient for the experienced demand, a phenomenon which is compounded by the highly variable nature (in both frequency and quantity) of spare part demand in aviation [1–3]. Due to this high variability, actual demand is difficult to estimate or forecast accurately. Consequently, this drives companies to keep relatively high stock buffers in order to ensure the availability of parts, leading to increased holding costs and waste of part life.

As noted by Regattieri et al. [2], many MROs and airlines do not use sophisticated techniques for demand estimation and forecasting, but rely on in-house experience or component supplier suggestions. If forecasting is in place, time-series techniques are often employed. Regattieri et al. [2] analysed the accuracy of twenty time-series forecasting techniques and noted their strengths and weaknesses. In a similar vein, Ghobbar and

Friend [4] stated that airline operators could improve their forecasts by identifying which drivers induce the variable behaviour of demand.

The latter points towards one of the noteworthy limitations in the current state of the art, which is that many studies do not provide further understanding of the generation of demand due to the use of time-series techniques, where demand is the only variable taken into account. Multiple demand drivers are not usually considered, either individually or on a joint basis. Furthermore, a substantial subset of the literature assumes that the state of installed components is always in a "as-good-as-new" state, i.e., denoting perfect repairs, or a "bad-as-old" state, i.e., denoting minimal repairs [5]. Neither assumption necessarily matches with spare part configurations in which overhauled or refurbished components are reintroduced into service. While the current state of the art does present models for imperfect repairs [6–10] and applies them for policy evaluation and optimisation purposes [11,12], their use for the prediction of demand behaviour has not been explored, to the best of the authors' knowledge.

One of the downsides of the use of time-series techniques is that no further understanding of the generation of the demand is provided. As the study of Van der Auweraer et al. [13] noted, installed base information can be used to forecast the upcoming demand of spare parts. Similarly, environmental conditions can be used to improve the quality of forecasts [14]. The research by Lowas and Ciarallo [15] uncovered some reasons for the unpredictable behaviour of spare part demand. The most significant single factor driving demand variability was found to be the size of the fleet of aircraft. It was concluded that smaller fleets have higher values for the Coefficient of Variance (CV) and Average Demand Interval (ADI)—measures of the variability in quantity and frequency of demand—when compared to large fleets. The authors of the study recommended further study to better understand demand generation drivers, as only some were tested.

One commonly made assumption throughout the literature is related to the state of the component when installed on an aircraft. Studies using the expected lifetime of components often neglect the fact that errors in the repair process occur and hence repairable components are not restored in an as-good-as-new state [6,7,16]. As maintenance personnel face high levels of time pressure and the effects of environmental circumstances in the industry, errors in the process will occur. Research has shown that in at least 39% of cases, maintenance errors are related to installation errors or incomplete repairs [17]. Broken components that are placed back into operation result in subsequent failures due to the incorrect state of the component [18]. This leads to concentrations in failures, resulting in a peak in spare part demand. However, this type of failure dependency is typically not addressed within the scope of spare part demand forecasting.

This research aims to address the aforementioned limitations in the state of the art by (1) modelling, simulating and evaluating the effect of incorrect repairs on demand patterns; and (2) quantifying the effect of multiple demand drivers in conjunction, leading to an improved understanding of demand driver priority. In terms of demand drivers, fleet size, incorrect repairs, environmental conditions, and different component commonality strategies are considered.

Section 2 gives an overview of the academic state of the art regarding the research topic, and further highlights limitations that will be addressed in this research. Section 3 presents the modelling and simulation approach. Subsequently, this approach is applied to a case study comprising real-life component data from an aircraft MRO provider that services CS25 category aircraft. Section 4 presents the case study characteristics and gives the results of a quantitative evaluation. Next, Section 5 discusses the validity and applicability of the results. Finally, Sections 6 and 7 present the conclusions and recommendations for future studies.

## 2. Literature Review

In this section, the state of the art with respect to spare part demand estimation and forecasting, as well as the modelling of dependent failures, is discussed. This includes

an identification of gaps, highlighting the associated research contributions of the current work.

### 2.1. Spare Part Demand Characterisation and Forecasting

For MRO providers, being able to accurately forecast spare part demand is crucial to optimally maintain aircraft. When spare part demand is more predictable, as expressed in lower levels of forecast error, inventory levels can be lowered. Currently, the aviation industry carries around EUR 30 billion yearly in spare parts stock to keep aircraft airworthy [19].

To classify spare part demand behaviour, a variety of metrics can be employed. Two metrics are prevalent: average inter-demand arrival (ADI); and the squared coefficient of variation ($CV^2$) [20]. Figure 1 visualises four resulting demand patterns that can be identified using these two metrics and their associated threshold values. Demand is typically characterised as intermittent for patterns that have infrequent demand with low variability in quantity (with cut-off values in the literature suggesting an ADI > 1.32 and a $CV^2$ < 0.49), and as lumpy for demand that is highly variable in both frequency and quantity (ADI > 1.32 and $CV^2$ > 0.49) [20,21].

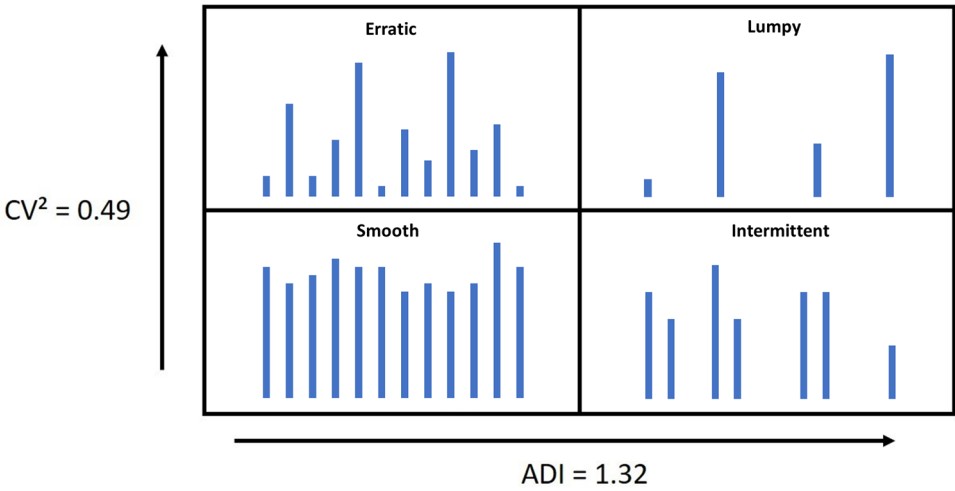

**Figure 1.** Overview of demand patterns based on ADI and $CV^2$.

As stated in Section 1, various studies have found that spare part demand in aviation tends to be intermittent or lumpy [1,4,14].

The research by Boone et al. [22], which focused on critical challenges related to inventory management in service parts supply, concluded that the inaccuracy of spare parts forecasts was the only challenge selected unanimously by the participating panel members. Furthermore, it was ranked as the second-most-difficult challenge facing spare parts inventory managers. Hence, intermittent and lumpy behaviour results in a problem for the inventory management of MRO providers.

Over the years, methods used for forecasting have been further developed and improved. Ghobbar and Friend [1] highlight various categories of methods used for forecasting, including trending methods, causal methods and time-series methods. Time-series methods range from relatively simple methods, such as moving averages and exponential smoothing variants, to more advanced methods, such as Croston's method [23]. In the latter method, the forecasts for the demand size and the demand interval are treated separately in order to minimise the error of the forecast. In 2005, Syntetos and Boylan [24] improved upon Croston's method by addressing the bias present in this method using an adjusted forecast factor $(1 - \alpha/2)$. Another improvement to Croston's method was realised in the research of Romeijnders et al. [25]. This research proved that two-step forecasting methods are more accurate than the benchmarked technique. The method showed a 20%

reduction in forecast error. In summary, the current literature is still working to improve the applicability and accuracy of various time-series techniques.

In recent years, causal methods have seen increased interest. As multiple factors influence the demand of different aircraft components, it is crucial for deeper understanding to identify these causalities and quantify their impact. Demand is generated when a component is removed from service, mostly due to failure or a strategic decision to replace the component.

At the aircraft level, the research of Ghobbar and Friend [4] showed the effects of aircraft utilisation and flight hours on the failures of components, as the wear and tear of components increased with increasing utilisation and flying hours, therefore increasing the demand rate for spare parts. When considering fleet-level demand, Low and Ciarallo [15] proved that small fleets have higher demand $CV^2$ and higher ADI than large fleets, and that higher buy periods tend to inflate the $CV^2$ and ADI as well. In both studies, a correlation between spare part demand and certain demand drivers was substantiated. However, both studies stated that there was still little understanding of the causes of fluctuations in spare part demand. Notably, to the best of the authors' knowledge, existing research into causal methods does not address the influence of the maintenance quality of the components. In other words, whether a component is correctly or incorrectly repaired is typically not taken into account in the existing demand forecasting literature. The state of the component upon installation is crucial for the lifetime—and therefore the moment of failure—of the component. As the intermittency of the demand of spare parts is one of the key aspects limiting the improvement of the predictability of demand, a better understanding of the drivers of demand-generating behaviour is needed. Incorrect repairs lead to components that do not function when put into service, triggering subsequent failures. Hence, this results in a demand peak for the components under consideration. Given this, the current research aims to address the associated gap in the state of the art regarding the potential influence of incorrect repairs on spare part demand.

### 2.2. Dependent Failures

Despite the goal of part M and part 145 organisations being to prevent the failure of components from happening while extending their lifetime for as long as possible within safety margins, failures do occur in aviation. Incorrect repairs cause components to remain in a broken state, having a crucial impact on the functioning both of the component itself and of interdependent components. When placed back into service, a broken component will not function as desired, and hence, interdependent components will also be affected. An example of this is the failure of components due to overloading; the given workload is fully dependent on the working components, as broken components cannot take any load, which may lead to subsequent overloading of the components that are still functioning. These phenomena are currently present in the aviation industry [17,26–28].

Various models can be used to represent imperfect or incorrect repairs [16,29,30]. Of these, the Branching Poisson Process [29] makes it possible not only to model incorrect repairs, but also to estimate the occurrence of subsequent dependent failures, making it highly suitable for application in the context of the stated research aim. According to the underlying theory, a primary failure might trigger subsequent failures, dependent on the correctness of the repair. Here, $r$ is the probability that a repair is not performed correctly. Hence, $1 - r$ represents the chance that the repair is executed correctly. In cases where the repair is performed incorrectly, the incorrect repair will spawn a finite renewal process of subsidiary failures. The number of subsidiary failures is a discrete random variable. At the time of the first primary failure $(Z_1) = z$, the expected number of failures in the interval $[0, t]$ can be expressed as $H(t - z)$. Then, the contribution of the first event in the subsidiary process for the expected events can be described as follows:

$$E\left[N^{(1)}(t)\right] = \left\{ E\left[N^{(1)}(t)\Big|Z_1\right] \right\} = \int_0^t E\left[N^{(1)}(t)\Big|Z_1 = z\right] f_1(z)dz = \int_0^t H(t-z)f_1(z)dz \tag{1}$$

where $f_1(z)$ represents the probability density function of the primary events. The same representation can be made for the expected number of failures $N^k(t)$ in $[0, t]$ due to the $k$th subsidiary process:

$$E\left[N^{(k)}(t)\right] = \int_0^t H(t-z)f_k(z)dz \qquad (2)$$

That stated, the expected number of failures of any type in $[0, t]$, $\Lambda(t)$, can be expressed as the sum of the expected number of primary failures in $[0, t]$ and the expected number of subsidiary failures. This leads to:

$$\Lambda(t) = E[N(t)] = \Lambda_z(t) + \int_0^t H(t-z)\Lambda_z'(z)dz \qquad (3)$$

A graphical overview of the occurrence of primary and subsidiary failures is provided in Figure 2. As can be seen from the figure, the complete process considers the superposition of the primary and subsidiary events of failures. Here, the assumption is made that the two types of events are indistinguishable.

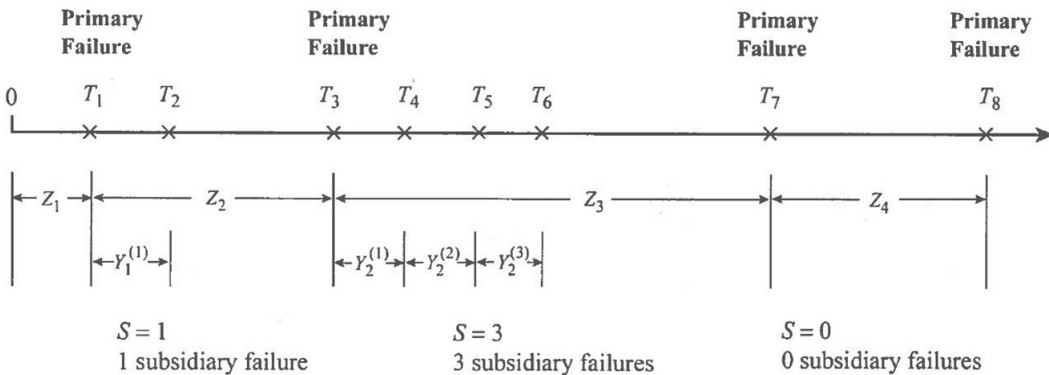

**Figure 2.** Visual presentation of the BPP [29].

## 3. Modelling and Simulation Approach

In this section, in which the model formulation and implementation are described, the required input data and the simulation setup are elaborated. Section 3.1 provides the explanation of the modelling and simulation approach, and Section 3.2 describes the approach towards a systematic evaluation of the influencing parameters.

### 3.1. Methods

The approach aims to provide a quantitative answer to what the impact of different levels of repair quality is on spare part demand. Therefore, it was decided to capture the impact on ADI, $CV^2$ and the overall number of failures when varying the levels of repair quality. The first two metrics cover the variability in demand, whereas the final metric captures the overall demand size.

In order to incorporate the effect of incorrect repairs on spare part demand in conjunction with several other drivers of demand, the developed approach incorporates a model for characterising incorrect repair in combination with a Monte Carlo simulation to generate spare part demand sequences on the basis of multiple input parameters. A visualisation of this approach is provided in Figure 3. As randomness is in play with the occurrence of incorrect repairs and the distribution of subsidiary failures, a total of 50 iterations of the model are performed before analysing the results. The final results are based on distributional characteristics taken from across the individual iterations.

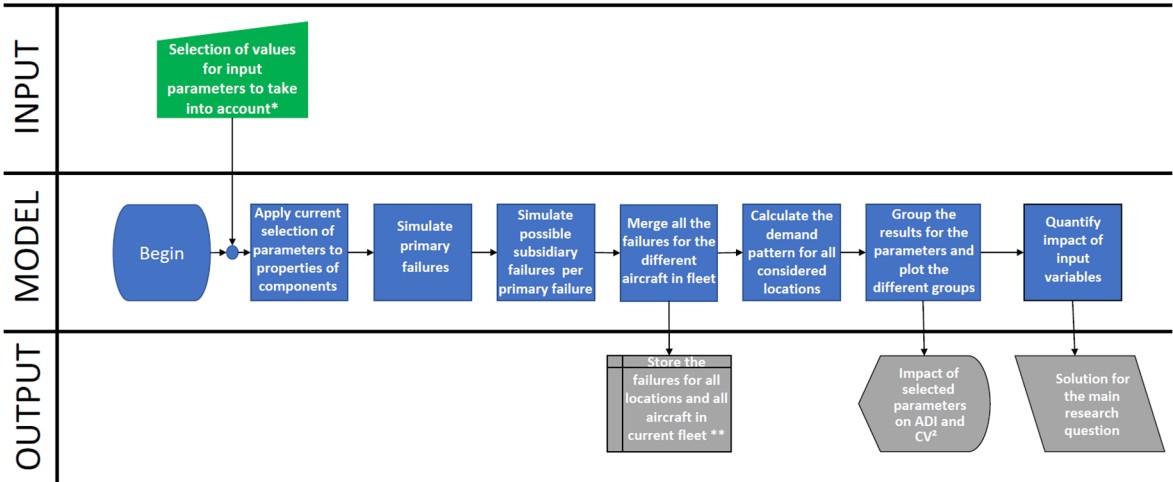

* Incorporates the different demand drivers that are taken into account as well as the values for lambda and r
** For every component, each failure in each aircraft is stored such that the time of the failure can be obtained later on

**Figure 3.** Model flowchart.

For modelling incorrect repairs, a Branching Poisson Process (BPP) is implemented. The BPP utilises a parameter, $r$, that represents the chance of an incorrect repair being performed. This parameter influences the discrete random variable that represents the spawning of subsequent failures. Hence, when an incorrect repair takes place and the component is placed back into service, the number of failures during a relatively short timespan may peak due to a certain number of subsequent failures. The influence of this parameter on $CV^2$ and ADI is the core attribute of this model. Although it could be argued that the value of $r$ might change over time, and that therefore a time-dependent function $r(t)$ might be present, this is not undertaken in this study.

Aside from the main parameter $r$, additional parameters that may influence spare part demand patterns are considered. The work of Lowas and Ciarallo [15] showed the influence of fleet size on demand patterns. Thijssens and Verhagen [14] showed that environmental conditions impact the reliability of components for multiple different reasons. Air pollutants and salinity all have an impact on the corrosion process of components. In addition to a natural reference climate (i.e., temperate), humid and desert climates are taken into account as well, both of which affect the Mean Time Between Failure (MTBF) (note that other relevant metrics include Mean Time Between Repair (MTBR) and Mean Time Between Overhaul (MTBO), but the cited study focuses on MTBF). The impact of incorrect repairs in combination with these other varying circumstances provides a wider perspective on the general behaviour of spare part demand.

The BPP and the previously highlighted parameters are implemented and subsequently simulated for every aircraft in the fleet. Failures at the selected aircraft component locations (expressed using system ATA codes; see Section 3.2) are simulated according to their corresponding failure rate $\lambda$, obtained from the analysis of the underlying data. Based on the number of primary removals that contain subsequent removals in a data set, an estimate of the probability of an incorrect repair $r$ can be made for the specific component location. Subsequently, possible subsequent failures are simulated. Next, the results of the individual aircraft are summed, resulting in the sum of the failures over time for every ATA location that is selected to be part of the model. This is done for multiple combinations of parameters. The values for ADI and $CV^2$ are stored. The above-mentioned metrics describe the predictability of the failures over time, but do not provide an answer regarding the quantity of failures. Therefore, this metric is added to the results as well, in order to capture both the behaviour as well as the sum of the failures.

The model can be applied across a variety of scenarios. In this study, four scenario variants are considered: each variant builds on the previous one to allow for the progressive

generation of results, enabling the evaluation of individual effects followed by joint effects. The variants and their progressive nature are briefly discussed below.

- Variant 1—Base: In this variant, the level of repair is the only parameter to be varied. Only temperate environmental conditions are taken into account. All fleet sizes are taken into account, but no distinction is made in the presentation of the results. No increase in component commonality across different aircraft is taken into consideration.
- Variant 2—Incorporation of varying fleet sizes: This variant uses the same set of results as Variant 1, but a distinction between the different fleet sizes is made in the presentation of the results.
- Variant 3—Incorporation of varying fleet sizes and environmental conditions: As different environmental conditions influence the effect of the expected lifetime of components, this will result in varying values for the different λs. Here, the results of humid and desert environments are also taken into account.
- Variant 4: Incorporation of varying fleet sizes, environmental conditions, and component commonality strategies. As flag carriers tend have more diverse fleets when compared to low-cost carriers [31], MRO providers have to deal with different aircraft types. Component commonality across the aircraft types is typically limited, with aircraft types in a family concept usually sharing the greatest degree of commonality. However, recent research by Zhang et al. [32] has shown promising results regarding potential gains with respect to costs when component commonality is increased. Hence, this variant investigates the effect of the increment of component commonality. From a practical perspective, this may give insights into any additional requirements on aircraft and component design, where OEMs have an opportunity to increase the similarity of components across multiple aircraft types. This has obvious manufacturing and supply chain benefits, but using Variant 4, it becomes possible to assess any potential effects on spare part demand.

The model algorithm can be described as per the pseudo-code given in Algorithm 1.

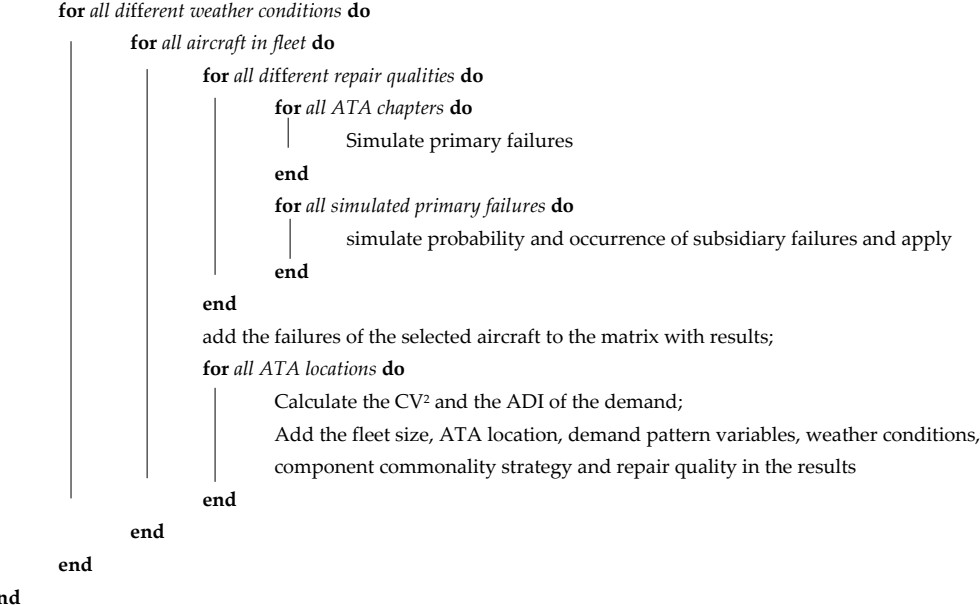

**Algorithm 1:** Pseudocode of model

---

**input:** All possible different values for *r*, weather conditions, fleet sizes and component commonality strategies
**output:** The $CV^2$ and ADI of the demands of spare parts and the total amount of failures
**for** *all different fleet sizes* **do**
　　**for** *all different weather conditions* **do**
　　　　**for** *all aircraft in fleet* **do**
　　　　　　**for** *all different repair qualities* **do**
　　　　　　　　**for** *all ATA chapters* **do**
　　　　　　　　　　Simulate primary failures
　　　　　　　　**end**
　　　　　　　　**for** *all simulated primary failures* **do**
　　　　　　　　　　simulate probability and occurrence of subsidiary failures and apply
　　　　　　　　**end**
　　　　　　**end**
　　　　　　add the failures of the selected aircraft to the matrix with results;
　　　　　　**for** *all ATA locations* **do**
　　　　　　　　Calculate the $CV^2$ and the ADI of the demand;
　　　　　　　　Add the fleet size, ATA location, demand pattern variables, weather conditions, component commonality strategy and repair quality in the results
　　　　　　**end**
　　　　**end**
　　**end**
**end**

*3.2. Parameters*

In order to determine the influence of the different scenarios, multiple parameter values have to be taken into consideration.

The main goal of this research is to reveal the impact of the quality of the repair process for components that are placed back into the aircraft on the $CV^2$ and ADI of spare part demand. As the initial values of *r* are retrieved from the data analysis of the dataset, these values are used as reference values (i.e., the Normal scenario). Scenarios with values for *r* increased by 100% (the Worse scenario), or decreased by 50% (the Improved scenario) or 100% (the Perfect scenario) are tested. The first of the mentioned alterations of *r* represents a scenario in which the amount of incorrectly repaired components that is placed back into service is twice as high as the reference scenario. The second alteration represents a scenario in which the chances of an incorrect repair are decreased by 50%. Therefore, less incorrectly repaired components are placed back into service. The last option represents a scenario where no incorrectly repaired components are placed back into the aircraft, and thus all components that are placed back function properly.

The work of Thijssens and Verhagen [14] showed the impact of three environmental factors on the Restricted Mean Survival Time (RMST) of components in aviation. The RMST is equal to the mean survival time, except that the RMST is restricted to within a time range $[0, \theta]$ to avoid the negative influences of the poorly determined right tail of a survival curve during estimation [33]. In this study, the impact of the environmental factors is directly related to the MTBF of components by the numerical factor provided in Table 1. For every aircraft considered in the analysis, the airline can be traced back via the external organisation code. In this way, the dominant environmental conditions at the main hub of the airline can be applied, and the values for the specific aircraft can be adjusted.

**Table 1.** The effect of environmental factors on the MTBF [14].

| Natural Climate | MTBF Ratio |
|:---:|:---:|
| Temperate | 1 |
| Humid | 0.62 |
| Desert | 0.73 |

The study by Lowas and Ciarallo [15] provided insights into the reasons for lumpy spare part demand. The study found that the parameter with the greatest impact on the lumpiness of the demand for spare parts was the fleet size. In order to validate this finding and to extend its scope, it is tested in this research, as well. As the reference study clearly described the range of values selected for the fleet size, this was not further thematised in this study, and the same range of values were chosen for the model. Finally, the increment of component commonality across different aircraft types was tested. Here, it is assumed that different aircraft types perform differently, resulting in variations in the average operating time of components. A deviation of 20% is assumed. The size of the deviation itself is not crucial, as the outcome will be directly compared to variants in which no different aircraft are considered. If significant differences are observed, this will serve as a stepping stone motivating the development of future research.

Table 2 represents the parameters discussed in the previous paragraph and used in the Monte Carlo simulation in a single consistent overview.

**Table 2.** Monte Carlo simulation parameters.

| Parameter | Tested Values | | | | | | | | Number of Steps |
|:---:|:---:|:---:|:---:|:---:|:---:|:---:|:---:|:---:|:---:|
| *r* factor | 0.0 | | 0.5 | | 1 | | 2 | | 4 |
| Environmental factors | | Natural | | | Humid | | Desert | | 3 |
| Fleet size | 8 | 16 | 32 | 64 | 96 | 128 | (256) | (512) | 6 (8) |
| Component commonality | | Not present | | | | Present | | | 2 |

## 4. Results

Before the application of the proposed approach and the subsequent evaluation of the results, this section starts with a brief discussion of the case study application and the associated data characteristics. Section 4.1 provides insights into the dataset and the manner in which input data for the model are generated.

### 4.1. Case Study Characteristics

In order to provide the model with the right input parameters based on the failure behaviour of aircraft components, data from an anonymous aircraft manufacturer were used. The data consist of removal data spanning across multiple decades.

For each data point, in this research, the part number, date, aircraft type, ATA chapter code (denoting the associated (sub)system), the serial number of the aircraft, and the operator are used. A selection of components is made in order to limit the scope. An overview of this analysis can be found in Table 3.

**Table 3.** Component trade-off at the ATA chapter level.

| ATA Chapter | System | Data Points | Occurrences with Subsidiary Failures | Average MTBR of Primary Failures [days] | Removals in All Aircraft | Total Unique Aircraft | Flight Safety-Critical | Selected |
|---|---|---|---|---|---|---|---|---|
| 21 | Air Conditioning | 2909 | 489 | 131 | Yes | 182 | No | No |
| 22 | AutoFlight | 2892 | 332 | 118 | Yes | 159 | No | No |
| 23 | Communications | 4053 | 595 | 131 | Yes | 195 | Yes | Yes |
| 24 | Electrical Power | 1697 | 397 | 60 | Yes | 192 | Yes | Yes |
| 25 | Equipment/Furnishings | 3483 | 342 | 159 | Yes | 166 | No | No |
| 26 | Fire Protection | 447 | 117 | 112 | Yes | 112 | No | No |
| 27 | Flight Controls | 1761 | 419 | 131 | Yes | 161 | Yes | Yes |
| 28 | Fuel | 1773 | 318 | 164 | Yes | 175 | Yes | Yes |
| 29 | Hydraulic Power | 1290 | 283 | 130 | Yes | 168 | Yes | Yes |
| 30 | Ice & Rain Protection | 1597 | 365 | 153 | Yes | 177 | Yes | Yes |
| 31 | Indicating/Recording System | 2002 | 431 | 106 | Yes | 185 | No | No |
| 32 | Landing Gear | 9195 | 448 | 127 | Yes | 217 | No | No |
| 33 | Lights | 2971 | 416 | 125 | Yes | 189 | Yes | Yes |
| 34 | Navigation | 8233 | 982 | 85 | Yes | 225 | No | No |
| 35 | Oxygen | 4355 | 294 | 155 | Yes | 176 | Yes | Yes |
| 36 | Pneumatic | 1110 | 184 | 125 | Yes | 170 | No | No |
| 38 | Water/Waste | 803 | 154 | 107 | Yes | 108 | No | No |
| 49 | Airborne Auxiliary Power | 1533 | 349 | 149 | No | 108 | No | No |
| 52 | Doors | 328 | 73 | 180 | Yes | 68 | No | No |
| 53 | Fuselage | 174 | 47 | 154 | No | 48 | No | No |
| 55 | Stabilizers | 157 | 41 | 193 | No | 41 | No | No |
| 56 | Windows | 666 | 122 | 143 | Yes | 122 | No | No |
| 57 | Wings | 304 | 60 | 99 | No | 59 | Yes | No |
| 61 | Propellers/Propulsion | 498 | 98 | 153 | No | 63 | Yes | No |
| 71 | Power Plant General | 345 | 87 | 127 | Yes | 91 | Yes | No |
| 72 | Engine Turbine/Turboprop, Ducted Fan/Unducted Fan | 256 | 64 | 123 | Yes | 72 | Yes | No |

**Table 3.** *Cont.*

| ATA Chapter | System | Data Points | Occurrences with Subsidiary Failures | Average MTBR of Primary Failures [days] | Removals in All Aircraft | Total Unique Aircraft | Flight Safety-Critical | Selected |
|---|---|---|---|---|---|---|---|---|
| 73 | Engine Fuel & Control | 868 | 191 | 112 | Yes | 135 | Yes | No |
| 74 | Ignition | 104 | 22 | 204 | No | 24 | Yes | No |
| 75 | Air | 142 | 33 | 133 | No | 34 | Yes | No |
| 75 | Engine Controls | 146 | 37 | 233 | Yes | 37 | Yes | No |
| 76 | Engine Indicating | 1088 | 193 | 149 | Yes | 160 | Yes | No |
| 77 | Exhaust | 399 | 107 | 252 | Yes | 107 | Yes | Yes |
| 78 | Oil | 161 | 34 | 156 | No | 35 | No | No |
| 79 | Starting | 575 | 132 | 107 | Yes | 132 | Yes | No |
| 80 | Airborne Auxiliary Power | 1773 | 318 | 164 | Yes | 175 | Yes | No |

This limits the scope to components in the following eight ATA chapters: 23 (Communications), 24 (Electrical Power), 27 (Flight Controls), 28 (Fuel), 29 (Hydraulic Power), 32 (Landing gear), 34 (Navigation) and 77 (Engine Indicating). Based on the operator, the environmental conditions can be determined for every aircraft in the data set. This has a direct impact on the lifetime of the components, and therefore influences the Mean Time Between Failure (MTBF), and hence the spawn rate of primary failures [14]. From the selected data, primary and subsidiary removals could be identified. In this analysis, a subsidiary removal is defined as a removal occurring within fourteen days of the primary removal. Here, it is assumed that components are interdependent if and only if they are located in the same ATA chapter. With this information, the spawn rate of primary removals can be determined for every ATA chapter code and location on every aircraft. Adjustments are made with respect to the environmental conditions in order to be able to correctly quantify the effect of variations in environmental conditions. With an overview of primary and subsidiary removals, the likelihood of an incorrect repair occurring at each ATA location can be made by reviewing the number of primary failures that incorporate subsidiary removals. For every location, the composition of subsequent failures is reviewed. Through this, in the proposed approach, the offset of a primary failure can be varied based on the distribution of the offset from the data. In implementation, the subsidiary failures are randomly distributed over the fourteen days following the day on which the primary failure occurs.

Next, for all aircraft and ATA locations, a check has to be made regarding the homogeneity of the primary removal rates of the components. The results of this test are presented in Table 4. It can be concluded that the spawn rate of primary removals is constant in most cases. Therefore, a Homogeneous Poisson Process can be used to simulate these removals. Furthermore, no significant differences were found among the performances of the different aircraft represented in the data. Hence, the results could be aggregated.

**Table 4.** Results of the trend analysis for the failure rate of the primary removal.

| No Trend | Increasing Trend | Decreasing Trend |
|---|---|---|
| 92.86% | 2.44% | 4.70% |

The results are presented in the order of the four different variants considered in this study. Visualisations of the results are available, although only a small selection of all visualisations are provided here for ease of interpretation. In the figures, each data point represents the average demand characteristics (ADI and $CV^2$) of a unique combination of varying parameters.

For each variant, the impact of improving the repair quality is quantified. The motivation for presenting the deviations resulting from this improvement originates from the desire of MRO providers to minimize the number of errors during the repair process and to strive for improvement. Therefore, MRO providers can use the outcomes of this study to quantify the effect of improving their repair quality on ADI, $CV^2$ and total number of failures.

The results of the statistical tes"s ar' provided in Appendix A. The results are presented in the form of *p*-values of the Mann–Whitney *U* test and the Kruskal–Wallis *H* test [34–36]. Values that are not significantly different according to these tests ($p > 0.05$), are marked with an asterisk in the tables.

### 4.2. Variant 1—The Influence of Variations in Repair Quality

The visual representation of the results in Figure 4 does not directly indicate a discernible difference in performance with different levels of repair quality. Tables 5 and 6 provide a quantitative comparison. Here, the comparison is made between the current level of repair quality (left column) and the desired level of repair quality (top row). The number provides a ratio of the average value of the metric of the desired level of repair quality and the current level of repair quality. Table 7 provides the total number of failures for the different levels of repair quality. Here, it can be seen that improved levels of repair quality result in lower numbers of failures.

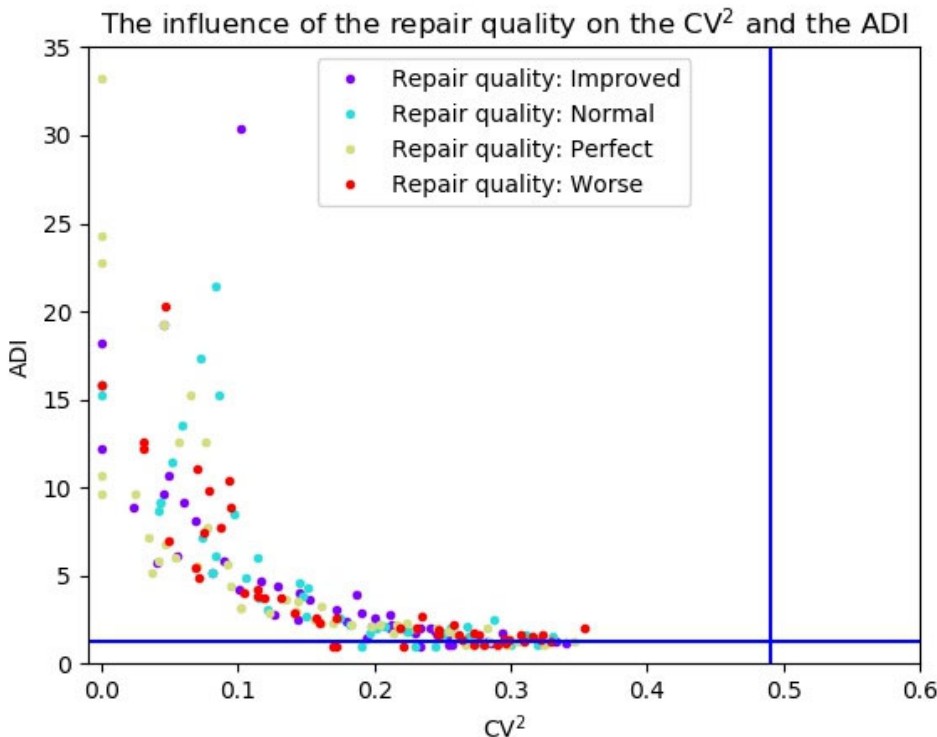

**Figure 4.** Results for Variant 1.

**Table 5.** Overview of the influence on the average ADI of varying the levels of repair quality.

| Repair Quality | Worse | Normal | Improved | Perfect |
|:---:|:---:|:---:|:---:|:---:|
| Worse | 1.000 | 1.151 | 1.278 | 1.462 |
| Normal | 0.869 | 1.000 | 1.126 | 1.283 |
| Improved | 0.782 | 0.888 | 1.000 | 1.152 |
| Perfect | 0.684 | 0.779 | 0.868 | 1.000 |

**Table 6.** Overview of the influence on the average $CV^2$ of varying the level of repair quality.

| Repair Quality | Worse | Normal | Improved | Perfect |
|---|---|---|---|---|
| Worse | 1.000 | 0.970 | 0.916 | 0.842 |
| Normal | 1.031 | 1.000 | 1.011 | 0.926 |
| Improved | 1.091 | 0.989 | 1.000 | 0.961 |
| Perfect | 1.187 | 1.079 | 1.041 | 1.000 |

**Table 7.** Average number of failures for the different levels of repair quality per iteration.

| Repair Quality | Worse | Normal | Improved | Perfect |
|---|---|---|---|---|
| Failures | 18,329 | 15,115 | 12,763 | 10,200 |

Improving the repair quality from "Worse" to "Normal" increased the ADI by 15.1%, decreased the $CV^2$ by 3.0% and decreased the total number of failures by 17.5%.

Improving the repair quality from "Normal" to "Improved" led to an increase in the ADI by 12.6%, an increase in the $CV^2$ by 1.1%, and a reduction in the total number of failures by 15.6%. Hence, this improvement has a positive effect on the total number of failures, but decreases the predictability of failures over time.

Improving the repair quality from the "Improved" level to the "Perfect" level resulted in an improvement in ADI by 15.2%, a reduction in $CV^2$ by 3.9%, and a reduction in the total number of failures by 20.1%.

### 4.3. Variant 2—The Influence of Variations in Repair Quality and Fleet Size

It can be seen from the results presented in Figure 5 that an increased fleet size lowers the ADI and increases the $CV^2$. The results of varying the repair quality for all fleet sizes are provided in Table 8.

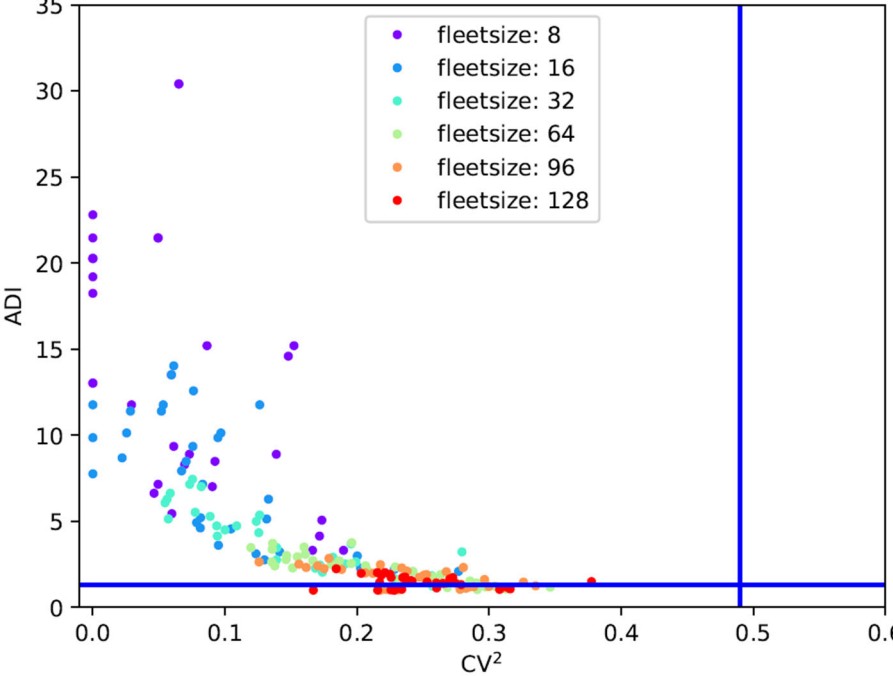

**Figure 5.** Results of Variant 2: influence of *r* on $CV^2$ and ADI for different fleetsizes.

**Table 8.** Quantitative overview of evaluation of Variant 2.

| Improvement | Fleet Size | ADI | CV$^2$ | Failures |
|---|---|---|---|---|
| Worse—Normal | 8 | +18.4% | −17.8% | −19.9% |
| | 16 | +13.2% | −29.7% | −17.2% |
| | 32 | +16.5% | −23.4% | −18.9% |
| | 64 | +11.2% | −9.2% | −17.7% |
| | 96 | +9.2% | −7.3% | −17.7% |
| | 128 | +7.8% | −2.2% | −18.5% |
| Normal—Improved | 8 | +16.4% | +3.9% | −13.0% |
| | 16 | +17.1% | +7.8% | −16.0% |
| | 32 | +12.1% | +0.7% | −15.2% |
| | 64 | +9.8% | −4.9% | −16.1% |
| | 96 | +6.8% | +0.5% | −15.2% |
| | 128 | +5.3% | −0.0% | −15.2% |
| Improved—Perfect | 8 | +18.6% | −10.8% | −21.4% |
| | 16 | +20.4% | −9.1% | −19.4% |
| | 32 | +14.5% | −16.3% | −19.8% |
| | 64 | +11.4% | −11.3% | −19.9% |
| | 96 | +10.8% | −6.9% | −20.7% |
| | 128 | +7.8% | −2.1% | −20.1% |

Generally, it can be concluded that improved repair quality results in a higher ADI, a lower CV$^2$ and a lower total number of failures. It can be seen from Table 8 that the impact of improvements in repair quality is larger with smaller fleet sizes. However, the impact on the number of failures is not strongly influenced by fleet size. Hence, it can be seen that this decrease remains somewhat constant for different fleet sizes.

It is interesting to note that the improvement in repair quality from "Normal" to "Improved" in most cases does not have a positive effect on CV$^2$—wee the italicised numbers in Table 8. This can be explained by the fact that, although fewer subsequent failures occur, the variance in demand quantity increases at a higher rate than the mean value of demand quantity. An example is given in Table 9. For each level of repair quality, an overview of the number of failures at each time point is provided. Primary failures are indicated by bold numbers, subsequent failures are provided as regular text. It can be observed that the time series for the "Improved" scenario has longer periods of zero demand, causing a more lumpy demand pattern when compared to the "Normal" and "Worse" scenarios. The CV$^2$ rises accordingly.

**Table 9.** Straightforward example of increasing CV$^2$ with better repair quality.

| | Worse | Normal | Improved |
|---|---|---|---|
| 1 | 1 | 1 | 1 |
| 2 | 1 | 0 | 0 |
| 3 | 2 | 0 | 0 |
| 4 | 0 | 0 | 0 |
| 5 | 1 | 1 | 1 |
| 6 | 2 | 2 | 0 |
| 7 | 2 | 2 | 0 |
| 8 | 0 | 0 | 0 |
| 9 | 1 | 1 | 1 |
| 10 | 2 | 2 | 2 |
| 11 | 0 | 0 | 0 |
| 12 | 2 | 2 | 2 |
| $\mu$ | 1.56 | 1.57 | 1.4 |
| $\sigma$ | 0.50 | 0.49 | 0.49 |
| CV$^2$ | 0.103 | 0.097 | 0.122 |

### 4.4. Variant 3—The Influence of Variations in Repair Quality, Fleet Size and Environmental Conditions

The combination of fleet size and climate is taken into account here, and eighteen different scenarios (six fleet sizes, three environmental conditions) were generated. However, these scenarios were increasingly hard to interpret. Hence, only a quantitative overview in the form of Table 10 is provided. The table provides the influence of changing the level of repair quality on ADI and $CV^2$. Note that the results of temperate environmental conditions were already provided in Section 4.3. The table shows the deviations of the improvement displayed in the first column.

**Table 10.** Quantitative overview of evaluation of Variant 3.

| Improvement | Fleet Size | ADI | | | CV² | | | Failures | | |
|---|---|---|---|---|---|---|---|---|---|---|
| | | Temperate | Humid | Desert | Temperate | Humid | Desert | Temperate | Humid | Desert |
| Worse— Normal | 8 | +18.4% | +13.9% | +13.9% | −17.8% | −22.7% | −18.8% | −19.9% | −16.7% | −16.8% |
| | 16 | +13.2% | +15.2% | +15.6% | −29.7% | −27.8% | −24.6% | −17.2% | −17.2% | −17.4% |
| | 32 | +16.5% | +12.8% | +13.5% | −23.4% | −10.7% | −15.7% | −18.9% | −16.8% | −16.9% |
| | 64 | +11.2% | +8.7% | +10.1% | −9.2% | −6.1% | −7.8% | −17.7% | −15.8% | −17.4% |
| | 96 | +9.2% | +6.1% | +7.8% | −7.3% | −1.0% | −1.0% | −17.7% | −16.0% | −17.2% |
| | 128 | +7.8% | +4.8% | +6.0% | −2.2% | +2.7% | +0.4% | −18.5% | −16.0% | −16.9% |
| Normal— Improved | 8 | +16.8% | +17.1% | +15.7% | +3.9% | +12.9% | +3.4% | −13.0% | −13.3% | −13.8% |
| | 16 | +17.1% | +10.8% | +13.6% | +7.8% | +2.8% | +6.0% | −16.0% | −12.3% | −13.7% |
| | 32 | +12.1% | +8.9% | +11.4% | +0.7% | −5.6% | −3.8% | −15.2% | −13.5% | −14.8% |
| | 64 | +9.8% | +6.9% | +7.2% | −4.9% | −2.0% | −2.8% | −16.1% | −14.0% | −13.4% |
| | 96 | +6.8% | +5.0% | +5.7% | +0.5% | +1.6% | −1.6% | −15.2% | −13.5% | −14.0% |
| | 128 | +5.3% | +3.6% | +4.0% | −0.0% | +4.4% | +3.4% | −15.2% | −13.1% | −14.2% |
| Improved— Perfect | 8 | +18.6% | +14.2% | +17.0% | −10.8% | −13.4% | −7.3% | −21.4% | −17.9% | −18.3% |
| | 16 | +20.4% | +16.2% | +15.5% | −9.1% | −13.7% | −10.4% | −19.4% | −18.6% | −19.1% |
| | 32 | +14.5% | +13.0% | +12.5% | −16.3% | −12.0% | −13.6% | −19.8% | −17.8% | −18.9% |
| | 64 | +11.4% | +8.0% | +9.8% | −11.3% | −4.3% | −6.4% | −19.9% | −17.8% | −19.4% |
| | 96 | +10.8% | +6.7% | +7.2% | −6.9% | −0.0% | −1.7% | −20.7% | −18.3% | −18.8% |
| | 128 | +7.8% | +4.7% | +5.8% | −2.1% | +3.5% | +1.2% | −20.1% | −18.1% | −18.7% |

The main addition of Variant 3 to the study is the exploration of the effect of improving the level of repair quality for varying environmental conditions, expressed in the values for ADI, $CV^2$ and total failures.

For desert and humid environments, patterns similar to those of temperate environmental conditions were found. The changes in ADI and $CV^2$ were dampened when the fleet size becomes larger, while the relative losses in total failures remained somewhat constant. Both the "Worse to Normal" and "Improved to Perfect" improvements performed similarly for all metrics. However, the improvement in repair quality from "Normal to Improved" saw a limited decrease in $CV^2$. In fact, many scenarios induce an increase in the $CV^2$. This is similar to the results of Variant 2.

By comparing the temperate and humid environmental scenarios, it can be concluded that under humid conditions, the ADI is less sensitive to the improvement in the repair quality. This results in smaller increments in ADI compared to under temperate environmental conditions. The results of the deviation in $CV^2$ provide no clear winner, as both environmental conditions outperform the other conditions for different values of fleet size and improvement. The relative losses in total failures are higher for temperate environmental conditions, although the difference between the two scenarios is small.

When comparing the temperate and desert environmental scenarios, it is clear that the desert environmental conditions perform slightly better than the temperate conditions when it comes to increasing the ADI. That is to say, the increase in the ADI under the same situation is slightly less compared to the increase in the ADI under temperate environmental

conditions. When comparing the deviations in $CV^2$, no clear pattern can be found. In some cases, desert conditions outperform the temperate conditions, but the opposite occurs for the same number of scenarios. With respect to decreasing the total number of failures, desert conditions are slightly less advantageous compared to temperate environmental conditions.

Generally, the increment in ADI with improvement is the most limited under humid conditions, the performance with decreasing $CV^2$ is similar for all environmental conditions, and the relative reduction in the total number of failures is similar for all environmental conditions, although the temperate environmental conditions perform slightly better in most cases.

### 4.5. Variant 4—The Influence of Variations in Repair Quality, Fleet Size, Environmental Conditions and Component Commonality Strategies

The individual results of the outcome of this variant are not presented, but are directly compared with the results of Variant 3. In this way, the impact of increasing the component commonality index can be evaluated. Hence, the results are discussed with the support of Table 11. Here, the results are presented as the difference in performance between Variant 3 and Variant 4. Therefore, if, in a certain scenario, the ADI of Variant 3 is increased by 1.0% and the ADI of Variant 4 is increased by 2.0%, the table will state a difference of +1.0%.

**Table 11.** Quantitative overview of evaluation of Variant 3.

| Improvement | Fleet Size | ADI | | | $CV^2$ | | | Failures | | |
|---|---|---|---|---|---|---|---|---|---|---|
| | | Temperate | Humid | Desert | Temperate | Humid | Desert | Temperate | Humid | Desert |
| Worse—Normal | 8 | −8.0% | +1.2% | −0.3% | −1.6% | −4.0% | −12.7% | +3.8% | −0.3% | +0.1% |
| | 16 | +3.7% | +2.1% | −0.8% | +1.6% | +1.1% | +6.9% | −1.7% | −0.9% | +0.4% |
| | 32 | −3.1% | −1.3% | −0.7% | +6.7% | −3.9% | −1.8% | +2.5% | +1.1% | −0.3% |
| | 64 | −0.6% | +0.5% | −0.2% | −1.0% | +1.1% | +0.2% | −0.1% | −0.6% | +0.5% |
| | 96 | +1.1% | +0.2% | +0.2% | +1.0% | +0.3% | −0.9% | −1.4% | −0.0% | −0.2% |
| | 128 | −0.4% | −0.3% | −0.2% | +0.1% | −0.0% | +2.3% | +0.2% | +0.1% | −0.2% |
| Normal—Improved | 8 | +5.4% | −1.3% | +4.9% | +1.9% | +5.0% | +1.3% | −3.0% | +0.5% | −2.9% |
| | 16 | +1.7% | +1.2% | +1.3% | −2.7% | −0.3% | −0.8% | −0.8% | −0.4% | −0.7% |
| | 32 | +0.7% | +2.3% | −1.3% | −6.6% | +0.4% | −1.2% | −0.3% | −0.2% | +0.7% |
| | 64 | −0.4% | −0.7% | +0.2% | +2.1% | −0.2% | −0.2% | +0.8% | +1.0% | +0.7% |
| | 96 | +0.3% | −0.2% | −0.8% | −2.6% | −0.7% | +0.6% | −0.0% | +0.3% | +0.5% |
| | 128 | +0.9% | +0.2% | +0.2% | −0.0% | −0.3% | −2.6% | −0.2% | −0.5% | −0.0% |
| Improved—Perfect | 8 | −5.2% | −1.6% | −6.5% | +2.4% | −5.3% | −1.3% | +3.4% | +1.7% | +2.8% |
| | 16 | −7.4% | +1.7% | −2.6% | −0.7% | −2.1% | −6.7% | +2.0% | +0.1% | +1.2% |
| | 32 | −0.2% | −0.4% | +0.5% | −1.7% | +1.1% | +4.4% | −0.0% | −0.3% | +0.3% |
| | 64 | +0.9% | +0.4% | −0.1% | +0.7% | +0.4% | −0.9% | −0.1% | +0.3% | +0.6% |
| | 96 | −1.5% | −0.2% | +0.7% | +0.8% | +0.1% | +0.6% | +0.3% | +0.6% | −0.2% |
| | 128 | −0.4% | −0.1% | +0.1% | −0.2% | −0.5% | +0.1% | −0.3% | +0.3% | +0.1% |

Although in some cases there are significant differences, most of the deviations are relatively small. Hence, the variability in spare part demand is not deteriorated by the introduction of components that are operable for multiple aircraft types.

## 5. Discussion

For different fleet sizes, employing shared component strategies among different aircraft types and environmental conditions, the influence of repair quality was quantified by capturing the changing values for the ADI and $CV^2$. Table 12 provides the total-effect indices obtained using Sobol's sensitivity analysis for the different variables [37].

**Table 12.** Sensitivity analysis: first- and total-effect indices.

|  |  | Fleet Size | Repair Quality | Component Commonality | Environment |
|---|---|---|---|---|---|
| S1 | $CV^2$ | 0.818 | 0.083 | 0.028 | 0.028 |
|  | ADI | 0.868 | 0.021 | 0.012 | 0.022 |
| ST | $CV^2$ | 0.885 | 0.136 | 0.078 | 0.090 |
|  | ADI | 0.942 | 0.052 | 0.049 | 0.063 |

The total-effect index translates the contribution to the output variance of the variable. The influence of the fleet size is dominant for the variance in the outcomes of both ADI and $CV^2$. Therefore, it can be concluded that the fleet size is the main influencing factor for both metrics. This suggests that adjusting the fleet size will have the greatest impact on potentially lowering the ADI and $CV^2$. However, for many reasons, the expansion of the fleet is not always possible. In cases where this expansion is not feasible and the fleet sizes cannot be increased, the influence of repair quality on the demand pattern becomes more dominant. This can be seen in Table 13, where the fleet size is fixed and the variance in the outcome depends on repair quality, component commonality, and environmental conditions.

**Table 13.** Sensitivity analysis: first- and total-effect indices.

|  |  | Repair Quality | Component Commonality | Environment |
|---|---|---|---|---|
| S1 | $CV^2$ | 0.427 | 0.182 | 0.204 |
|  | ADI | 0.293 | 0.286 | 0.367 |
| ST | $CV^2$ | 0.636 | 0.361 | 0.413 |
|  | ADI | 0.354 | 0.339 | 0.440 |

A critical note has to be made regarding the values of parameters for different levels of repair quality. The results obtained from the data analysis are used as a reference scenario (i.e., the "Normal" repair quality), while the other three are based on a multiplication of this scenario. The values of the parameters for different levels of repair quality were chosen in order to conduct a thorough numerical evaluation. In practice, the difference in performance is unlikely to be of this size.

The strong assumption regarding the interdependency among components in the same ATA chapter results in the limitation of the usefulness of the outcome when it comes to the location of failures and the corresponding failure patterns. In other words, while the ATA chapter results are related to aircraft systems, and an assumption of interdependency is made, this is not necessarily true. The same ATA code can refer to multiple instances of a system on a single aircraft; for example, a failure related to ATA chapter 38, which covers waste/water systems, may in fact relate to different instances of galleys, bathrooms, wastewater tanks, etc., which are located at a number of different places in the aircraft, and may not have interdependent functionality. In addition, components could be connected with and dependent on components belonging to different ATA chapters. Without detailed ATA chapter indications (using the full six digits to describe systems at the unit level) or additional information on the location (for instance, through ATA zonal codes), conclusions with respect to failure location and their associated patterns are difficult to draw, and spare part demand can only be determined at a higher level of system aggregation. From this perspective, when aggregating the results of the failures, the locations of the components are not decisive for the outcomes of this research.

Another assumption is made in Variant 4, where the effect of the different component commonalities in a fleet is tested. Due to the lack of research and data on component commonality across heterogeneous fleets, only a rough estimation of the associated effect on component demand was performed.

As this research quantifies the impact of repair quality on the different demand metrics, repair quality is used as a varying parameter in the model. However, the influence of changes in other varying parameters might also affect the metrics. Hence, there is no proof that all changes are the result of variations in repair quality alone, and variances caused by the interaction of the different parameters should also be included. Table 12 provides the first- and total-effect indices of the sensitivity analysis. As can be seen from the table, the differences among the first- and total-effect indices are relatively small. Hence, the influence of the interaction is limited.

Another important note should be made regarding the statistical outcomes of the Kruskal–Wallis *H* and Mann–Whitney *U* tests. As the commonly chosen 95% interval provides a fair threshold for the rejection of the null hypothesis, *p*-values below 0.05 cause the null hypothesis to be rejected and thus it can be assumed that different groups of data have different medians. However, this *p*-value is highly dependent on the number of data points in the compared groups. As the number of iterations for the simulation was set to 50, the sizes of the subsets grew by a factor 50. Therefore, the *p*-values became smaller, resulting in a more frequent rejection of the null hypothesis. However, when reviewing only a single iteration, the *p*-values are higher, and the null hypothesis is rejected less often. It is, however, not an option to exclude the iterations from the model, as these iterations provide outcome stability by omitting the random factor.

A final critical note can be made on the limited set of drivers for failures. As frequently stated in previous research, not all drivers of failure are known, resulting in research that includes a limited number of drivers. However, this research provides a broadening to the current knowledge by including the effect of different levels of repair quality.

## 6. Conclusions

The impact of changing repair quality on the predictability of the failures of components was quantified. In general, the following can be established:

- An improvement in repair quality induces an increase in ADI, a reduction in $CV^2$ and a reduction in the total number of failures.
- For larger fleet sizes (more than 64 aircraft of the same type), the effects of increased repair quality on the ADI and $CV^2$ become less significant, while the effect on the total failures remains the same.

Therefore, it can be concluded that when facing larger fleets, the improvement in repair quality has a wider support base, as the downside of the implementation becomes smaller. Ironically, larger fleets have fewer problems with variability in spare part demand.

This research contributes towards a more complete understanding of the way in which drivers for component spare part demand may behave. To the best of the authors' knowledge, this study is the first to explicitly address the influence of repair quality on the demand behaviour of components, while systematically exploring and verifying the influence of a range of additional demand drivers. This gives further insight into spare part demand behaviour under more realistic conditions, where multiple drivers may apply at the same time.

## 7. Recommendations and Future Research

Based on the presented research, the following recommendations can be identified:

- Future research investigating the variance in the repair quality among different MRO providers is advised in order to verify the assumptions included in the presented approach.
- Furthermore, not only incorrect repairs, which leave the component in the same broken state as before, but minimal and imperfect repairs, for instance as represented in the Kijima Type I or II models, can be considered, as well [30]. This would enable a more realistic representation of the repair process, instead of the current somewhat simplified representations available in spare part demand driver literature.

- Regarding practical considerations, should the aviation industry decide to further investigate the possibilities regarding component pooling across multiple aircraft types, a detailed analysis of the performance of individual aircraft types would lead to a more accurate prediction of the performance of a heterogeneous fleet compared to representation as a homogeneous fleet.
- Finally, future research may address and reveal interdependencies among flight-safety-critical components for different types of aircraft. The results of such a study would contribute to the practical relevance regarding the patterns and locations of primary and subsidiary failures over time.

**Author Contributions:** Conceptualisation, L.M.H. and W.J.C.V.; methodology, L.M.H. and W.J.C.V.; validation, L.M.H.; formal analysis, L.M.H.; investigation, L.M.H.; resources, L.M.H. and W.J.C.V.; data curation, L.M.H.; writing—original draft preparation, L.M.H.; writing—review and editing, W.J.C.V.; visualisation, L.M.H.; supervision, W.J.C.V.; project administration, W.J.C.V. All authors have read and agreed to the published version of the manuscript.

**Funding:** This research received no external funding.

**Data Availability Statement:** Restrictions apply to the availability of these data. Data was obtained from a commercial maintenance, repair and overhaul (MRO) provider and are available from the authors with the permission of the MRO provider.

**Conflicts of Interest:** The authors declare no conflict of interest.

## Appendix A. Statistical Test Results

**Table A1.** Overview of the p-values of the Mann-Whitney U-test for ADI of the different scenarios.

| Repair Quality | Worse | Normal | Improved | Perfect |
|---|---|---|---|---|
| Worse | 0.500 | 0.008 | 0.000 | 0.000 |
| Normal | 0.008 | 0.500 | 0.029 | 0.000 |
| Improved | 0.000 | 0.029 | 0.500 | 0.015 |
| Perfect | 0.000 | 0.000 | 0.015 | 0.500 |

**Table A2.** Overview of the p-values of the Mann-Whitney U-test for $CV^2$ of the different scenarios.

| Repair Quality | Worse | Normal | Improved | Perfect |
|---|---|---|---|---|
| Worse | 0.500 | 0.010 | 0.000 | 0.000 |
| Normal | 0.010 | 0.500 | 0.056 | 0.000 |
| Improved | 0.000 | 0.056 | 0.500 | 0.003 |
| Perfect | 0.000 | 0.000 | 0.003 | 0.500 |

**Table A3.** Outcome of the *p*-values of the Mann-Whitney *U*-tests.

| Fleet Size | Repair Quality | ADI | | | | $CV^2$ | | | |
|---|---|---|---|---|---|---|---|---|---|
| | | Worse | Normal | Improved | Perfect | Worse | Normal | Improved | Perfect |
| 8 | Worse | 0.500 | 0.000 | 0.000 | 0.000 | 0.500 | 0.000 | 0.000 | 0.000 |
| | Normal | | 0.500 | 0.062 | 0.000 | | 0.500 | 0.081 | 0.000 |
| | Improved | | | 0.500 | 0.000 | | | 0.500 | 0.000 |
| | Perfect | | | | 0.500 | | | | 0.500 |
| 16 | Worse | 0.500 | 0.000 | 0.000 | 0.000 | 0.500 | 0.003 | 0.000 | 0.000 |
| | Normal | | 0.500 | 0.000 | 0.000 | | 0.500 | 0.007 | 0.000 |
| | Improved | | | 0.500 | 0.000 | | | 0.500 | 0.000 |
| | Perfect | | | | 0.500 | | | | 0.500 |

**Table A3.** *Cont.*

| Fleet Size | Repair Quality | ADI Worse | Normal | Improved | Perfect | CV² Worse | Normal | Improved | Perfect |
|---|---|---|---|---|---|---|---|---|---|
| **32** | Worse | 0.500 | 0.000 | 0.000 | 0.000 | 0.500 | 0.000 | 0.000 | 0.000 |
| | Normal | | 0.500 | 0.000 | 0.000 | | 0.500 | 0.008 | 0.000 |
| | Improved | | | 0.500 | 0.000 | | | 0.500 | 0.000 |
| | Perfect | | | | 0.500 | | | | 0.500 |
| **64** | Worse | 0.500 | 0.000 | 0.000 | 0.000 | 0.500 | 0.000 | 0.000 | 0.000 |
| | Normal | | 0.500 | 0.000 | 0.000 | | 0.500 | 0.000 | 0.000 |
| | Improved | | | 0.500 | 0.000 | | | 0.500 | 0.000 |
| | Perfect | | | | 0.500 | | | | 0.500 |
| **96** | Worse | 0.500 | 0.000 | 0.000 | 0.000 | 0.500 | 0.000 | 0.000 | 0.000 |
| | Normal | | 0.500 | 0.000 | 0.000 | | 0.500 | 0.289 | 0.000 |
| | Improved | | | 0.500 | 0.000 | | | 0.500 | 0.000 |
| | Perfect | | | | 0.500 | | | | 0.500 |
| **128** | Worse | 0.500 | 0.000 | 0.000 | 0.000 | 0.500 | 0.010 | 0.000 | 0.000 |
| | Normal | | 0.500 | 0.000 | 0.000 | | 0.500 | 0.021 | 0.000 |
| | Improved | | | 0.500 | 0.000 | | | 0.500 | 0.002 |
| | Perfect | | | | 0.500 | | | | 0.500 |

**Table A4.** Outcome of the *p*-values of the Kruskal-Wallis *H*-test.

| Fleet Size | 8 | 16 | 32 | 64 | 96 | 128 |
|---|---|---|---|---|---|---|
| **ADI** | 0.000 | 0.000 | 0.000 | 0.000 | 0.000 | 0.000 |
| **CV²** | 0.000 | 0.000 | 0.000 | 0.000 | 0.000 | 0.000 |

**Table A5.** Outcome of the *p*-values of the Mann-Whitney U-tests for humid conditions.

| Fleet Size | Repair Quality | ADI Worse | Normal | Improved | Perfect | CV² Worse | Normal | Improved | Perfect |
|---|---|---|---|---|---|---|---|---|---|
| **8** | Worse | 0.500 | 0.000 | 0.000 | 0.000 | 0.500 | 0.007 | 0.000 | 0.000 |
| | Normal | | 0.500 | 0.000 | 0.000 | | 0.500 | 0.005 | 0.000 |
| | Improved | | | 0.500 | 0.000 | | | 0.500 | 0.002 |
| | Perfect | | | | 0.500 | | | | 0.500 |
| **16** | Worse | 0.500 | 0.000 | 0.000 | 0.000 | 0.500 | 0.000 | 0.000 | 0.000 |
| | Normal | | 0.500 | 0.001 | 0.000 | | 0.500 | 0.004 | 0.000 |
| | Improved | | | 0.500 | 0.000 | | | 0.500 | 0.001 |
| | Perfect | | | | 0.500 | | | | 0.500 |
| **32** | Worse | 0.500 | 0.000 | 0.000 | 0.000 | 0.500 | 0.001 | 0.000 | 0.000 |
| | Normal | | 0.500 | 0.000 | 0.000 | | 0.500 | 0.000 | 0.000 |
| | Improved | | | 0.500 | 0.000 | | | 0.500 | 0.000 |
| | Perfect | | | | 0.500 | | | | 0.500 |
| **64** | Worse | 0.500 | 0.000 | 0.000 | 0.000 | 0.500 | 0.000 | 0.000 | 0.000 |
| | Normal | | 0.500 | 0.000 | 0.000 | | 0.500 | 0.013 | 0.000 |
| | Improved | | | 0.500 | 0.000 | | | 0.500 | 0.001 |
| | Perfect | | | | 0.500 | | | | 0.500 |
| **96** | Worse | 0.500 | 0.000 | 0.000 | 0.000 | 0.500 | 0.006 | 0.001 | 0.000 |
| | Normal | | 0.500 | 0.000 | 0.000 | | 0.500 | 0.262 | 0.001 |
| | Improved | | | 0.500 | 0.000 | | | 0.500 | 0.002 |
| | Perfect | | | | 0.500 | | | | 0.500 |
| **128** | Worse | 0.500 | 0.000 | 0.000 | 0.000 | 0.500 | 0.382 | 0.288 | 0.390 |
| | Normal | | 0.500 | 0.000 | 0.000 | | 0.500 | 0.303 | 0.271 |
| | Improved | | | 0.500 | 0.000 | | | 0.500 | 0.149 |
| | Perfect | | | | 0.500 | | | | 0.500 |

**Table A6.** Outcome of the *p*-values of the Mann-Whitney U-tests for desert conditions.

| Fleet Size | Repair Quality | ADI | | | | CV$^2$ | | | |
|---|---|---|---|---|---|---|---|---|---|
| | | Worse | Normal | Improved | Perfect | Worse | Normal | Improved | Perfect |
| 8 | Worse | 0.500 | 0.000 | 0.000 | 0.000 | 0.500 | 0.000 | 0.000 | 0.000 |
| | Normal | | 0.500 | 0.004 | 0.000 | | 0.500 | 0.049 | 0.000 |
| | Improved | | | 0.500 | 0.000 | | | 0.500 | 0.001 |
| | Perfect | | | | 0.500 | | | | 0.500 |
| 16 | Worse | 0.500 | 0.000 | 0.000 | 0.000 | 0.500 | 0.003 | 0.000 | 0.000 |
| | Normal | | 0.500 | 0.000 | 0.000 | | 0.500 | 0.001 | 0.000 |
| | Improved | | | 0.500 | 0.000 | | | 0.500 | 0.001 |
| | Perfect | | | | 0.500 | | | | 0.500 |
| 32 | Worse | 0.500 | 0.000 | 0.000 | 0.000 | 0.500 | 0.000 | 0.000 | 0.000 |
| | Normal | | 0.500 | 0.000 | 0.000 | | 0.500 | 0.001 | 0.000 |
| | Improved | | | 0.500 | 0.000 | | | 0.500 | 0.000 |
| | Perfect | | | | 0.500 | | | | 0.500 |
| 64 | Worse | 0.500 | 0.000 | 0.000 | 0.000 | 0.500 | 0.000 | 0.000 | 0.000 |
| | Normal | | 0.500 | 0.000 | 0.000 | | 0.500 | 0.010 | 0.000 |
| | Improved | | | 0.500 | 0.000 | | | 0.500 | 0.000 |
| | Perfect | | | | 0.500 | | | | 0.500 |
| 96 | Worse | 0.500 | 0.000 | 0.000 | 0.000 | 0.500 | 0.267 | 0.000 | 0.000 |
| | Normal | | 0.500 | 0.000 | 0.000 | | 0.500 | 0.000 | 0.000 |
| | Improved | | | 0.500 | 0.000 | | | 0.500 | 0.008 |
| | Perfect | | | | 0.500 | | | | 0.500 |
| 128 | Worse | 0.500 | 0.000 | 0.000 | 0.000 | 0.500 | 0.111 | 0.076 | 0.000 |
| | Normal | | 0.500 | 0.000 | 0.000 | | 0.500 | 0.426 | 0.004 |
| | Improved | | | 0.500 | 0.000 | | | 0.500 | 0.002 |
| | Perfect | | | | 0.500 | | | | 0.500 |

**Table A7.** Outcome of the *p*-values of the Kruskal-Wallis *H*-test for humid conditions.

| Fleet Size | 8 | 16 | 32 | 64 | 96 | 128 |
|---|---|---|---|---|---|---|
| ADI | 0.000 | 0.000 | 0.000 | 0.000 | 0.000 | 0.000 |
| CV$^2$ | 0.000 | 0.000 | 0.000 | 0.000 | 0.000 | 0.780 |

**Table A8.** Outcome of the *p*-values of the Kruskal-Wallis *H*-test for desert conditions.

| Fleet Size | 8 | 16 | 32 | 64 | 96 | 128 |
|---|---|---|---|---|---|---|
| ADI | 0.000 | 0.000 | 0.000 | 0.000 | 0.000 | 0.000 |
| CV$^2$ | 0.038 | 0.002 | 0.000 | 0.007 | 0.000 | 0.000 |

**Table A9.** Outcome of the *p*-values of the Mann-Whitney U-tests for for temperate conditions and mixed fleet composition.

| Fleet Size | Repair Quality | ADI | | | | CV$^2$ | | | |
|---|---|---|---|---|---|---|---|---|---|
| | | Worse | Normal | Improved | Perfect | Worse | Normal | Improved | Perfect |
| 8 | Worse | 0.500 | 0.000 | 0.000 | 0.000 | 0.500 | 0.000 | 0.000 | 0.000 |
| | Normal | | 0.500 | 0.001 | 0.000 | | 0.500 | 0.011 | 0.000 |
| | Improved | | | 0.500 | 0.001 | | | 0.500 | 0.001 |
| | Perfect | | | | 0.500 | | | | 0.500 |
| 16 | Worse | 0.500 | 0.000 | 0.000 | 0.000 | 0.500 | 0.000 | 0.000 | 0.000 |
| | Normal | | 0.500 | 0.000 | 0.000 | | 0.500 | 0.003 | 0.000 |
| | Improved | | | 0.500 | 0.000 | | | 0.500 | 0.000 |
| | Perfect | | | | 0.500 | | | | 0.500 |

**Table A9.** *Cont.*

| Fleet Size | Repair Quality | ADI | | | | CV² | | | |
| --- | --- | --- | --- | --- | --- | --- | --- | --- | --- |
| | | Worse | Normal | Improved | Perfect | Worse | Normal | Improved | Perfect |
| 32 | Worse | 0.500 | 0.000 | 0.000 | 0.000 | 0.500 | 0.004 | 0.000 | 0.000 |
| | Normal | | 0.500 | 0.000 | 0.000 | | 0.500 | 0.000 | 0.000 |
| | Improved | | | 0.500 | 0.000 | | | 0.500 | 0.000 |
| | Perfect | | | | 0.500 | | | | 0.500 |
| 64 | Worse | 0.500 | 0.000 | 0.000 | 0.000 | 0.500 | 0.000 | 0.000 | 0.000 |
| | Normal | | 0.500 | 0.000 | 0.000 | | 0.500 | 0.005 | 0.000 |
| | Improved | | | 0.500 | 0.000 | | | 0.500 | 0.000 |
| | Perfect | | | | 0.500 | | | | 0.500 |
| 96 | Worse | 0.500 | 0.000 | 0.000 | 0.000 | 0.500 | 0.000 | 0.000 | 0.000 |
| | Normal | | 0.500 | 0.000 | 0.000 | | 0.500 | 0.014 | 0.000 |
| | Improved | | | 0.500 | 0.000 | | | 0.500 | 0.000 |
| | Perfect | | | | 0.500 | | | | 0.500 |
| 128 | Worse | 0.500 | 0.000 | 0.000 | 0.000 | 0.500 | 0.024 | 0.000 | 0.000 |
| | Normal | | 0.500 | 0.000 | 0.000 | | 0.500 | 0.033 | 0.000 |
| | Improved | | | 0.500 | 0.000 | | | 0.500 | 0.001 |
| | Perfect | | | | 0.500 | | | | 0.500 |

**Table A10.** Outcome of the *p*-values of the Mann-Whitney U-tests for for humid conditions and mixed fleet composition.

| Fleet Size | Repair Quality | ADI | | | | CV² | | | |
| --- | --- | --- | --- | --- | --- | --- | --- | --- | --- |
| | | Worse | Normal | Improved | Perfect | Worse | Normal | Improved | Perfect |
| 8 | Worse | 0.500 | 0.000 | 0.000 | 0.000 | 0.500 | 0.000 | 0.000 | 0.000 |
| | Normal | | 0.500 | 0.001 | 0.000 | | 0.500 | 0.093 | 0.000 |
| | Improved | | | 0.500 | 0.001 | | | 0.500 | 0.000 |
| | Perfect | | | | 0.500 | | | | 0.500 |
| 16 | Worse | 0.500 | 0.000 | 0.000 | 0.000 | 0.500 | 0.000 | 0.000 | 0.000 |
| | Normal | | 0.500 | 0.000 | 0.000 | | 0.500 | 0.000 | 0.000 |
| | Improved | | | 0.500 | 0.000 | | | 0.500 | 0.000 |
| | Perfect | | | | 0.500 | | | | 0.500 |
| 32 | Worse | 0.500 | 0.000 | 0.000 | 0.000 | 0.500 | 0.000 | 0.000 | 0.000 |
| | Normal | | 0.500 | 0.000 | 0.000 | | 0.500 | 0.000 | 0.000 |
| | Improved | | | 0.500 | 0.000 | | | 0.500 | 0.000 |
| | Perfect | | | | 0.500 | | | | 0.500 |
| 64 | Worse | 0.500 | 0.000 | 0.000 | 0.000 | 0.500 | 0.003 | 0.000 | 0.000 |
| | Normal | | 0.500 | 0.000 | 0.000 | | 0.500 | 0.011 | 0.000 |
| | Improved | | | 0.500 | 0.000 | | | 0.500 | 0.002 |
| | Perfect | | | | 0.500 | | | | 0.500 |
| 96 | Worse | 0.500 | 0.000 | 0.000 | 0.000 | 0.500 | 0.116 | 0.001 | 0.000 |
| | Normal | | 0.500 | 0.000 | 0.000 | | 0.500 | 0.031 | 0.000 |
| | Improved | | | 0.500 | 0.000 | | | 0.500 | 0.033 |
| | Perfect | | | | 0.500 | | | | 0.500 |
| 128 | Worse | 0.500 | 0.000 | 0.000 | 0.000 | 0.500 | 0.417 | 0.479 | 0.106 |
| | Normal | | 0.500 | 0.000 | 0.000 | | 0.500 | 0.465 | 0.060 |
| | Improved | | | 0.500 | 0.000 | | | 0.500 | 0.073 |
| | Perfect | | | | 0.500 | | | | 0.500 |

**Table A11.** Outcome of the *p*-values of the Mann-Whitney U-tests for for desert conditions and mixed fleet composition.

| Fleet Size | Repair Quality | ADI | | | | CV$^2$ | | | |
|---|---|---|---|---|---|---|---|---|---|
| | | Worse | Normal | Improved | Perfect | Worse | Normal | Improved | Perfect |
| 8 | Worse | 0.500 | 0.000 | 0.000 | 0.000 | 0.500 | 0.001 | 0.000 | 0.000 |
| | Normal | | 0.500 | 0.000 | 0.000 | | 0.500 | 0.024 | 0.000 |
| | Improved | | | 0.500 | 0.006 | | | 0.500 | 0.005 |
| | Perfect | | | | 0.500 | | | | 0.500 |
| 16 | Worse | 0.500 | 0.000 | 0.000 | 0.000 | 0.500 | 0.000 | 0.000 | 0.000 |
| | Normal | | 0.500 | 0.000 | 0.000 | | 0.500 | 0.012 | 0.000 |
| | Improved | | | 0.500 | 0.000 | | | 0.500 | 0.000 |
| | Perfect | | | | 0.500 | | | | 0.500 |
| 32 | Worse | 0.500 | 0.000 | 0.000 | 0.000 | 0.500 | 0.000 | 0.000 | 0.000 |
| | Normal | | 0.500 | 0.000 | 0.000 | | 0.500 | 0.000 | 0.000 |
| | Improved | | | 0.500 | 0.000 | | | 0.500 | 0.001 |
| | Perfect | | | | 0.500 | | | | 0.500 |
| 64 | Worse | 0.500 | 0.000 | 0.000 | 0.000 | 0.500 | 0.000 | 0.000 | 0.000 |
| | Normal | | 0.500 | 0.000 | 0.000 | | 0.500 | 0.005 | 0.000 |
| | Improved | | | 0.500 | 0.000 | | | 0.500 | 0.000 |
| | Perfect | | | | 0.500 | | | | 0.500 |
| 96 | Worse | 0.500 | 0.000 | 0.000 | 0.000 | 0.500 | 0.076 | 0.000 | 0.000 |
| | Normal | | 0.500 | 0.000 | 0.000 | | 0.500 | 0.004 | 0.000 |
| | Improved | | | 0.500 | 0.000 | | | 0.500 | 0.019 |
| | Perfect | | | | 0.500 | | | | 0.500 |
| 128 | Worse | 0.500 | 0.000 | 0.000 | 0.000 | 0.500 | 0.388 | 0.009 | 0.000 |
| | Normal | | 0.500 | 0.000 | 0.000 | | 0.500 | 0.004 | 0.000 |
| | Improved | | | 0.500 | 0.000 | | | 0.500 | 0.008 |
| | Perfect | | | | 0.500 | | | | 0.500 |

**Table A12.** Outcome of the *p*-values of the Kruskal-Wallis *H*-test for normal conditions and mixed fleet composition.

| Fleet Size | 8 | 16 | 32 | 64 | 96 | 128 |
|---|---|---|---|---|---|---|
| ADI | 0.000 | 0.000 | 0.000 | 0.000 | 0.000 | 0.000 |
| CV$^2$ | 0.000 | 0.000 | 0.000 | 0.000 | 0.000 | 0.000 |

**Table A13.** Outcome of the *p*-values of the Kruskal-Wallis *H*-test for humid conditions and mixed fleet composition.

| Fleet Size | 8 | 16 | 32 | 64 | 96 | 128 |
|---|---|---|---|---|---|---|
| ADI | 0.000 | 0.000 | 0.000 | 0.000 | 0.000 | 0.000 |
| CV$^2$ | 0.000 | 0.000 | 0.000 | 0.000 | 0.000 | 0.381 |

**Table A14.** Outcome of the *p*-values of the Kruskal-Wallis *H*-test for desert conditions and mixed fleet composition.

| Fleet Size | 8 | 16 | 32 | 64 | 96 | 128 |
|---|---|---|---|---|---|---|
| ADI | 0.001 | 0.000 | 0.000 | 0.000 | 0.000 | 0.000 |
| CV$^2$ | 0.001 | 0.000 | 0.000 | 0.002 | 0.000 | 0.000 |

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
