# Peer review of "The Influence of Repair Quality on Aircraft Spare Part Demand Variability"

_aerospace, doi:10.3390/aerospace10080731_

Round 1

Reviewer 1 Report

Dear Editor;

The manuscript titled “The Influence of Repair Quality on Aircraft Spare Parts Demand Variability” is evaluated meticulously from an academic perspective. In general, the article may be published upon your approval. However, there are some correction-required items as follows;

Comment - 1

General

The article provides information about MRO (Aircraft Approved Maintenance Organizations). It is also called the Part 145 company. However, it should be remembered that, an MRO organization cannot make the maintenance operations by itself. A Continuing Airworthiness Management Organization (Part M) should prepare the task cards, especially for the planned maintenance operations.

The whole manuscript should be revised considering the Part M organizations regarding scheduled/planned maintenance activities.

Comment - 2

(Line-93)

The abbreviation of MRO was used in line 43. With this respect, in Line 93, only MRO should be used.

Comment – 3

(Line-169-177)

The above-given paragraph provides very crucial information. However, there is only one reference [17]. Moreover, the reference was published in 2000. It is too old.

In this manner, more and updated references should be cited for the mentioned paragraph.

Comment - 4

Figure 3.

The letters are not readable. They should be rewritten in a larger format.

Comment – 5

(Line-276)

Not really. Vice-versa. Nowadays, flag carriers look at the same aircraft family for the benefit of similar maintenance operations and common parts storage.

Please check this argument and support it with more references.

Comment – 6

Figure 4 is too small to recognize. It should be bigger.

Comment – 7

(Line-242, 243)

Mean Time Between Failure (MTBF) was considered for the analysis. Nevertheless, two crucial concepts exist in the aviation maintenance industry, especially for commercial passenger aircraft. These are Mean Time Between Repair (MTBR) and Mean Time Between Overhaul (MTBO).

Comment – 8

Table 3 is too small to recognize. It should be prepared in a bigger size.

Comment – 9

Table 10 and Table 11 need to be more significant to recognize. It should be prepared in a bigger size.

Comment – 10

ATA Chapter is an appropriate categorization. However, the aircraft categorization also be given in the article. Is the topic related to CS23, or CS25?

Best Regards

Author Response

Comment 1

General

The article provides information about MRO (Aircraft Approved Maintenance Organizations). It is also called the Part 145 company. However, it should be remembered that, an MRO organization cannot make the maintenance operations by itself. A Continuing Airworthiness Management Organization (Part M) should prepare the task cards, especially for the planned maintenance operations.

The whole manuscript should be revised considering the Part M organizations regarding scheduled/planned maintenance activities.

Response: Thank you for this comment. We have clarified this throughout the manuscript, with respective updates in line 36-38 as well as highlighting the same point in line 170.

Comment – 2

The abbreviation of MRO was used in line 43. With this respect, in Line 93, only MRO should be used.

Response: Thank you for this comment. We have updated line 93 (currently line 96 in the version with tracked changes) to only feature the acronym (MRO), as it is introduced in full in line 35.  

Comment – 3

The above-given paragraph provides very crucial information. However, there is only one reference [17]. Moreover, the reference was published in 2000. It is too old. In this manner, more and updated references should be cited for the mentioned paragraph.

Response: this is an excellent catch. We agree this paragraph should be supported by more recent research. After reviewing several papers, we have included the following to support the points observed in this paragraph, with the reference numbering updated accordingly:

  1. Prasanna Illankoon, Phillip Tretten, Uday Kumar, A prospective study of maintenance deviations using HFACS-ME, International Journal of Industrial Ergonomics, Volume 74, 2019, 102852, ISSN 0169-8141, https://doi.org/10.1016/j.ergon.2019.102852.
  2. Jakub Kraus, Andrej Lališ, Vladimír Plos, Peter Vittek, Slobodan Stojić, Utilizing Ontologies and Structural Conceptual Models for Safety Data Management in Aviation Maintenance, Repair and Overhaul Organizations, Transportation Research Procedia, Volume 35, 2018, Pages 35-43, ISSN 2352-1465, https://doi.org/10.1016/j.trpro.2018.12.005.
  3. Miller, M., Mrusek, B. (2019). The REPAIRER Reporting System for Integrating Human Factors into SMS in Aviation Maintenance. In: Arezes, P. (eds) Advances in Safety Management and Human Factors. AHFE 2018. Advances in Intelligent Systems and Computing, vol 791. Springer, Cham. https://doi.org/10.1007/978-3-319-94589-7_44

Comment – 4

Figure 3. The letters are not readable. They should be rewritten in a larger format.

Response: we have increased the figure size and hope this resolves the issue of readability. If not successful, the font size of the letters can be increased but this will require a figure redesign to fit the used shapes; we are of course willing to do this, but perhaps the increased figure size will already do the job satisfactorily.

Comment – 5

Not really. Vice-versa. Nowadays, flag carriers look at the same aircraft family for the benefit of similar maintenance operations and common parts storage.

Please check this argument and support it with more references.

Response: this is correct. We rather meant to indicate that flag carriers, when compared to LCCs, are characterised by fleets that are typically more diverse, though undoubtedly the trend is as you indicate. We have adapted the associated sentence, which is now on line 279, and a further adjustment on line 282.

Comment – 6

Figure 4 is too small to recognize. It should be bigger.

Response: we have increased the size of the figure to increase readability. A further increase could be achieved by having the figure span the entire page, but we propose to defer to MDPI’s style advice during an eventual proofreading stage to indicate the best (allowable) option.

Comment – 7

Mean Time Between Failure (MTBF) was considered for the analysis. Nevertheless, two crucial concepts exist in the aviation maintenance industry, especially for commercial passenger aircraft. These are Mean Time Between Repair (MTBR) and Mean Time Between Overhaul (MTBO).

Response: this is correct. When considering unscheduled occurrences, Mean Time Between Unscheduled Removals (MTBUR) is another concept / metric of interest. Nonetheless, the analysis is based on factors from a reference (prior work from one of the study authors) where only MTBF was considered. As a solution, we have added a brief footnote on the other available metrics on page 6.

Comment – 8

Table 3 is too small to recognize. It should be prepared in a bigger size.

Response: similar to comment 6, we have increased the size of the table to increase readability. A further increase could be achieved by having the table span the entire page and/or rotate the table by 90 degrees and feature it on its own dedicated page, but we propose to defer to MDPI’s style advice during an eventual proofreading stage to indicate the best (allowable) option.

Comment – 9

Table 10 and Table 11 need to be more significant to recognize. It should be prepared in a bigger size.

Response: similar to comment 8, we have increased the size of these tables to increase readability. A further increase could be achieved by having the table span the entire page and/or rotate the table by 90 degrees and feature it on its own dedicated page, but we propose to defer to MDPI’s style advice during an eventual proofreading stage to indicate the best (allowable) option.

Comment – 10

ATA Chapter is an appropriate categorization. However, the aircraft categorization also be given in the article. Is the topic related to CS23, or CS25?

Response: the topic relates to CS25. We have added a note to this effect in line 95-96, see tracked changes.

Reviewer 2 Report

The topic of the article is very interesting one. And it is true that spare parts demand is difficult to estimate or forecast accurately. It will be very exciting if the method in this paper can obtain long-term tracking and iteration of MRO.

With regard to table 9, it is suggested that further explanations be added to make it more direct and clear.

Author Response

Thank you for your positive impressions and your helpful suggestion regarding table 9. To address this, additional explanation has been added in lines 436-439, see tracked changes and represented in full here: “It can be observed that the time series for the "Improved" scenario has longer periods of zero demand, causing a more lumpy demand pattern when compared to the "Normal" and "Worse" scenarios. The CV2 rises accordingly.“